# Hormone Receptor Loss in Breast Cancer: Molecular Mechanisms, Clinical Settings, and Therapeutic Implications

**DOI:** 10.3390/cells9122644

**Published:** 2020-12-09

**Authors:** Emma Zattarin, Rita Leporati, Francesca Ligorio, Riccardo Lobefaro, Andrea Vingiani, Giancarlo Pruneri, Claudio Vernieri

**Affiliations:** 1Fondazione IRCCS Istituto Nazionale dei Tumori, Via G. Venezian 1, 20133 Milan, Italy; emma.zattarin@istitutotumori.mi.it (E.Z.); rita.leporati@istitutotumori.mi.it (R.L.); francesca.ligorio@istitutotumori.mi.it (F.L.); riccardo.lobefaro@istitutotumori.mi.it (R.L.); andrea.vingiani@istitutotumori.mi.it (A.V.); giancarlo.pruneri@istitutotumori.mi.it (G.P.); 2Department of Oncology and Haematology, University of Milan, Via Festa del Perdono 7, 20122 Milan, Italy; 3IFOM, The FIRC Institute of Molecular Oncology, Via Adamello 16, 20139 Milan, Italy

**Keywords:** breast cancer, hormone receptors, conversion, intratumor heterogeneity, clonal selection, endocrine therapies, (neo)adjuvant therapy, tumor recurrences, re-characterization

## Abstract

Hormone receptor-positive breast cancer (HR+ BC) accounts for approximately 75% of new BC diagnoses. Despite the undisputable progresses obtained in the treatment of HR+ BC in recent years, primary or acquired resistance to endocrine therapies still represents a clinically relevant issue, and is largely responsible for disease recurrence after curative surgery, as well as for disease progression in the metastatic setting. Among the mechanisms causing primary or acquired resistance to endocrine therapies is the loss of estrogen/progesterone receptor expression, which could make BC cells independent of estrogen stimulation and, consequently, resistant to estrogen deprivation or the pharmacological inhibition of estrogen receptors. This review aims at discussing the molecular mechanisms and the clinical implications of HR loss as a result of the therapies used in the neoadjuvant setting or for the treatment of advanced disease in HR+ BC patients.

## 1. Introduction

Estrogen receptor alpha-positive (ERα+) breast cancer (BC) accounts for approximately 75% of all BC diagnoses [1]. About half of all ERα+ BCs also express the progesterone receptor (PgR), whose gene is under the transcriptional control of ERα and its ligands (e.g., estradiol or E2) [2]. ERα+/PgR+ BC is also referred to as hormone receptor (HR)-positive BC. 

By binding and activating ERα, E2 regulates the growth and differentiation of both normal mammalian cells and HR+ BC cells. Once ERα binds to estrogens, it homodimerizes and activates genomic and non-genomic signaling pathways that stimulate cancer cell growth and proliferation [3]. ERβ is a second isoform of ER, with distinct biological roles when compared with ERα. The effect of ERβ on the regulation of the gene expression has been less investigated, but several reports support its role in counteracting the ERα-induced stimulation of BC cell proliferation [4,5,6]. ERα and Erβ can also heterodimerize, consistent with the role of ERβ in modulating ERα activity [6,7]. HR status is a well-established prognostic and predictive factor in BC patients. Among the different BC subgroups, HR+ BC is characterized by lower growth and proliferation rates of cancer cells, as well as by longer disease-free survival (DFS) and overall survival (OS) after surgery, in addition to better OS in the metastatic disease setting. Even more importantly, HR expression is a predictive biomarker of the clinical benefit of endocrine therapies (ETs) in both limited-stage and advanced disease settings. Indeed, the assessment of the ERα/PgR expression by immunohistochemistry (IHC) is of the outmost importance in order to guide the choice of pharmacological therapies in all disease settings [8]. On the other hand, HR+ BC is generally less sensitive to cytotoxic chemotherapy, including neoadjuvant chemotherapy (NAC), when compared with HER2-positive (HER2+) BC and triple-negative breast cancer (TNBC) [9,10].

As HR+ BC depends on E2-induced or constitutive ERα activation for the stimulation of tumor cell growth and proliferation, different ETs resulting in the inhibition of ERα signaling have proven to be effective in the treatment of HR+ BC patients in both adjuvant and advanced disease settings. In particular, the selective estrogen receptor modulator (SERM) tamoxifen; the selective estrogen receptor downregulator (SERD) fulvestrant; or compounds that interfere with estrogen synthesis, such as aromatase inhibitors or ovarian suppression through luteinizing hormone-releasing hormone (LHRH) agonists, have become standard-of-care therapies in patients with HR+ BC in all clinical settings [11]. Despite the efficacy of these ETs, primary or acquired resistance of HR+ BC cells to ETs is a common event, especially in the advanced disease setting, where it is largely responsible for the failure of available ETs and, ultimately, for the death of the majority of BC patients. 

Several molecular mechanisms have been proposed to mediate primary or acquired resistance to ETs in HR+ BC. These mechanisms include mutations in the *ESR1* gene, which encodes ERα, or alterations in the genes of the mitogen activated protein kinase (MAPK) pathway (e.g., activating *ERBB2* mutations and *NF1* loss-of-function mutations) or *ESR1* transcriptional regulators, such as MYC (human homolog of the avian viral myelocytomatosis gene) [12]. Interestingly, some of these genomic mechanisms of resistance could be targeted by selective inhibitors that are already available for clinical use in other contexts [13]. Another potential mechanism of resistance to ETs in HR+ BC consists of the loss of ERαexpression, which could make cancer cells independent of E2− and ERα− induced stimulation of tumor cell growth and proliferation. While several studies have characterized the genomic mechanisms of HR+ BC cell resistance to ETs and their clinical consequences, the molecular mechanisms underneath HR conversion from HR+ to HR-negative (HR−) status, as well as their impact on patient prognosis and on the choice of therapeutic strategies, remain much less explored. This manuscript aims at reviewing and discussing (a) the available evidence regarding the molecular mechanisms that are responsible for HR status conversion in HR+ BC patients, (b) the clinical scenarios in which this phenomenon can occur, and (c) the therapeutic implications of HR status conversion in different clinical settings. 

## 2. Molecular Mechanisms Underlying Loss of HR Expression in BC Cells

It has been proposed that most ERα− negative (ERα−) BCs arise from ERα+ BC cells that subsequently loose ERα expression. Therefore, the identification of the mechanisms responsible for the changes in HR status could be helpful in the discovery of potentially effective therapeutic strategies to target ERα-BC cells [14] (Figure 1).

One major limitation of the molecular studies assessing HR status conversion is the fact that this phenomenon can occur in different phases of BC progression and, as a consequence, it could be driven by different mechanisms [15,16]. In particular, it remains unclear whether HR status conversion occurs spontaneously, or as a result of the selective pressure of estrogen deprivation in the patient blood or in the tumor microenvironment, or, finally, as an adaptive response to specific pharmacological treatments. Of note, not only estrogen deprivation or anti-estrogen compounds, but also cytotoxic agents, have been shown to modulate HR expression in BC cells [17,18]. 

### 2.1. Genetic Mechanisms

*ESR1* gene is located on chromosome 6q25.1, and spans eight exons and two non-coding regions over more than 140 Kb of DNA. ERα is a nuclear receptor protein, whose structure includes the N-terminal activation function 1 (AF1) domain; the DNA-binding domain (DBD) with two zinc finger motifs; the C-terminal domains, namely the 12-helical ligand-binding domain (LBD), which interacts with estrogens, and the ligand-dependent activation function 2 (AF2) domain, which is responsible for ligand-dependent Erα transactivation, including the cofactor-binding groove to which cofactors are recruited when the ERα becomes activated; and the flexible hinge domain, or D region, which contains the nuclear localization signal (NLS) and links the C-domain to the AF2 domain [19]. Two forms of Erα exist: the estrogen-occupied form, which is associated with nuclear chromatin and stimulates gene transcription, and the unoccupied form, which is loosely associated with nuclear chromatin and is, therefore, more easily extracted from the nuclei during hypotonic lysis of the cells [20,21].

When E2 binds ERα, it stimulates ERα homodimerization and binding to specific DNA regions known as estrogen responsive elements (EREs) within the enhancers and promoters of target genes (Figure 1) [22]. At the same time, AF-1 and AF-2 domains recruit coregulators that the remodel chromatin structure [23]. Through these mechanisms, ERα homodimers activate the transcription of the genes implicated in cell growth, proliferation, and signaling. 

When stimulated by E2-independent signals, such as AF1 phosphorylation by growth factor-dependent kinases or the palmitoylation of Cys^447^in the LBD, ERα also sustains intracellular molecular signaling through non-genomic mechanisms [24]. In fact, ERα localization is dynamic, and ERα shuttles between the nucleus and the cytoplasm, but it can also be detected in the cancer cells’ plasma membrane, where it directly interacts with several oncogenic proteins and stimulates oncogenic signaling pathways. Within the plasma membrane, ERα may form homodimers or couple with other proteins, such as caveolins or flotillins in the lipid rafts [25], where ERα transactivates the epidermal growth factor receptor (EGFR), HER2 receptor, insulin-like growth factor receptor I (IGFR I), the p85 regulatory subunit of phosphatidylinositol 3-kinase (PI3K), and G-proteins, relaying downstream proliferative and survival signals via mitogen-activated protein kinase (MAPK) and AKT [26].

Several studies have explored genetic *ESR1* alterations, such as gene mutations resulting in a constitutively active ERα, as a cause of endocrine resistance of HR+ BC cells; on the other hand, genetic alterations leading to a loss of ERα expression have been much less investigated [27]. Interestingly, while the loss of heterozygosity (LOH) of *ESR1* occurs in 18% of BCs, the most commonly detected *ESR1* alterations, i.e., gene deletions, insertions, rearrangements, or polymorphisms, are not commonly associated with a loss of ERα expression/activity. In particular, only a few studies support the development of BC resistance to ETs as a result of a lack in ERα function induced by the homozygous deletion of the *ESR1* gene or LOH with inactivating mutations in the remaining allele [28,29,30].

The *PGR* gene is located on chromosome 11q22-23, and spans eight exons over more than 90 kb [31]. It exists in two isoforms, A and B, which originate from two distinct transcription initiation sites and bear different biological functions; while PgR-A is a transcriptional repressor of both ERα and PgR-B, the latter acts as a transcription activator. Therefore, while PgR expression is induced by ERα, PgR binds ERα to modulate its signaling [32]. As in the case of the *ESR1* gene, *PGR* is also rarely mutated in primary BC [33]; in addition, the correlation between *PGR* mutations and the loss of PgR protein expression remains uncertain. On the other hand, LOH of the *PGR* gene occurs in approximately 18–40% of ERα+ BCs [34], and it has been associated with a loss of PgR protein expression [34].

### 2.2. Epigenetic Mechanisms

*ESR1* and *PGR* promoters are subjected to cyclic methylation/demethylation of CpG dinucleotides, which, in turn, can modulate ERα and PgR levels and the biological signaling downstream of ERα/PgR in human HR+ BC cells [35,36,37,38]. Upon *ESR1* promoter methylation, several transcription factors, such as AP2, cannot be recruited to the *ESR1* DNA locus any longer, thus resulting in the inhibition of *ESR1* transcription (Figure 1) [38]. Zinc-finger E-box binding homebox 1 (ZEB1), an important member of the zinc-finger-homeodomain transcription factor family, has been shown to repress *ESR1* transcription by forming a ZEB1/DNA methyltransferase (DNMT)3B/histone deacetylase 1 (HDAC 1) complex on the *ESR1* promoter, finally leading to its hypermethylation (Figure 1) [39]. Similarly, an increase in histone deacetylation has been shown to limit *ESR1* transcription by condensing the nucleosome structure [37].

In HR+ BC cells, the loss of ERα-mediated signaling also results in the repression of *PGR* gene transcription, a process that involves the recruitment of polycomb repressors and histone deacetylases to the *PGR* gene promoter, as well as *PGR* promoter methylation (Figure 2) [40]. Interestingly, one work showed that three methylation-sensitive restriction sites in the *PGR* gene CpG islands are not methylated in normal breast and in PgR+ BC specimens, but they are hypermethylated in approximately 40% of PgR− human BCs. These data suggest that the hypermethylation of *PGR* gene CpG islands is associated with a lack of PgR gene expression in a significant fraction of human BCs [41].

### 2.3. Growth Factor Signaling 

Several studies have shown a negative correlation between ERα expression/activation and the activation status of the PI3K/AKT/mTORC1 and MAPK signaling cascades. In the study by Perren A. et al., loss of phosphatase and tensin homolog (PTEN), an inhibitor of the PI3K/AKT/mTORC1 axis, was correlated with the loss of both ERα and PgR expressions in primary BC specimens (Figure 1), while Garcia J.M. et al. showed that LOH of the PTEN gene, which occurs in approximately 30–40% of sporadic BCs, is associated with a higher tumor histologic grade and loss of PgR, but not of ERα expression [42,43]. Moreover, short-term treatment with insulin-like growth factor-I (IGF-I), epidermal growth factor (EGF), and heregulin, induced the downregulation of *PGR* mRNA and PgR protein levels in BC cell lines [44]. 

Oh A.S. et al. showed that the hyperactivation of MAPK resulting from the overexpression of EGFR or c-erbB-2 results in ERα down-regulation (1) [45]. In particular, EGFR expression in human BC cell lines and in human BC specimens is inversely correlated with ERα expression, mainly as a result of the ERα-induced down-regulation of *EGFR* mRNA levels [46]. *HER2/neu* overexpression and HER2-mediated signaling have also been associated with the down-regulation of ERα and PgR expression. Approximately half of HER2+ BCs express ERα, whereas only ~10% of ERα+ BCs show HER2 overexpression [47]. HER2/neu overexpression/aberrant activation causes HR downregulation through both the PI3K/AKT/mTORC1 pathway, e.g., via AKT-mediated reduction of FOXO3a protein expression, and through the p42/44 MAPK pathway [45,46,47,48]. This could partly explain why ERα+/HER2+ BCs tend to express lower ERα levels when compared with ERα+/HER2 negative BCs, and they more frequently show relative resistance to ETs in the absence of concomitant HER2 inhibition [49]. Consistently, Creighton C.J. et al. showed that ERα+ PgR− BC cell lines overexpressing EGFR or constitutively active erbB-2 or RAF have a significantly lower expression of the *ESR1* gene [50]. A MAPK gene expression signature obtained from BC cell lines and validated in mRNA profiles of human BC datasets confirmed that MAPK hyperactivation profiles are enriched in ERα− BCs. Interestingly, the MAPK pathway-mediated downregulation of ERα expression has been demonstrated to be reversible and dynamic, and MAPK pathway inhibition results in ERα re-expression in MCF-7 cells.

Nuclear factor-kB complex (NFkB), a ubiquitously expressed family of inducible transcription factors, can also contribute to ERα down-regulation (Figure 1). In ERα- BC cell lines, NFkB is often found to be constitutively active, while its activation status is intermediate in ERα+/HER2+ BC cell lines; finally, a low or absent NFkB expression is typically found in ERα+/HER2 negative human BC cell lines [51]. In 81 primary human BC tissue samples, a higher NFkB activity is significantly correlated with lower ERα expression levels [52]. NFkB enhanced activation in ERα- BCs could be a consequence of MAPK hyperactivation [53] or, alternatively, it could depend on the PI3K/AKT/mTORC1 signaling pathway [54]. 

These data are consistent with the observation that the aberrant activation of growth factor signaling, including the MAPK and PI3K/AKT/mTORC1 pathways, is a common finding in HR− BC [55].

### 2.4. Post-Transcriptional Regulation of ERαExpression

*ESR1* gene transcription results in the synthesis of a mRNA molecule of 4.3 kb in length, with an extensive 3′ untranslated region (UTR) containing several regulatory elements, including long tracts of AU-rich sequences and 13 copies of AUUUA, which play a role in its destabilization through deadenylation and polyadenylase tail digestion [56]. Therefore, modifications of the *ESR1* mRNA sequence and structure could affect the ribosomal translation of ERα, but the mechanisms responsible for this phenomenon have not been fully elucidated yet [57].

Different microRNAs (miRNAs) interfere with *ESR1* mRNA translation (Figure 1) and, as a consequence, with ERα and ERβ levels. For instance, miR-222/221, which targets *ESR1* mRNA for degradation, has been found to be significantly higher in ERα− cells than in ERα+ BC cells [58]. In addition, miR-206 has also been found to bind the 3′ UTR of *ESR1* mRNA, while miR-92 targets the 3′ UTR of ERβ1 mRNA [59]. Finally, in the MCF-7 HR+ BC cell lines, miR-27a targets ZBTB10, a specificity protein that directly regulates ERα expression [60].

### 2.5. Post-Translational Regulation of ERαExpression

ERα activity can also be affected by post-translational modifications, including ubiquitination and phosphorylation. Interestingly, while E2 binds to ERα and stimulates its homodimerization and activation, it also promotes ERα polyubiquitination and degradation via the ubiquitin proteasome system (UPS). In addition, the binding of the E2-ERα complex to the EREs of the target genes leads to the recruitment of both coactivators of gene transcription and E3-ubiquitin ligases, thus balancing the E2-ERα complex transcriptional activity and ERα degradation/inhibition [61]. These regulatory mechanisms, which give rise to an incoherent feed-forward loop, self-limit the E2-induced stimulation of ERα, and result in a fine modulation of the intensity and duration of E2-ERα signaling. Of note, ERα polyubiquitination-mediated degradation is exploited by selective endocrine therapies acting as SERDs, such as fulvestrant, and new oral molecules that are being investigated in HR+/HER2− BC patients with tumors that are refractory to standard ETs (Figure 1) [62]. It is important to note that, while SERDs directly induce ERα degradation, they do not directly suppress the ERα transcriptional activity, which is instead driven by the induction of a conformational change in the receptor that disrupts the integrity of the primary coactivator binding surface and competitively displaces estradiol [63,64]. Therefore, both intracellular signals and pharmacological treatments can affect the intracellular ERα levels and biological activity.

ERα phosphorylation in different amino acidic residues modulates ERα ubiquitination. In BC cells grown in E2-containing media, SRC-induced phosphorylation of ERα in the Y537 residue induces the recruitment of E3-ubiquitin ligases, and the subsequent polyubiquitination and proteasome-mediated degradation of ERα [65]. Conversely, ERα phosphorylation by GSK3, LMTK3, and ABL stabilizes ERα and promotes its activity as a transcription factor (Figure 1) [61]. Other proteins and mechanisms converge on ERα degradation via UPS. For instance, Mucin 1 (MUC1) acts as an ERα coactivator by stabilizing its binding to ERE promoters and by recruiting histone acetyl-transferases of the p160 family to promote ERα-induced gene transcription [66]; consistently, MUC1-knock down in HR+ BC cells is associated with reduced ERα levels [61]. In addition, peptidyl prolyl isomerase1 (PIN1) prevents the interaction of the E3 ligase E6AP with ERα and, as a consequence, it inhibits ERα degradation [67]. The tumor suppressor retinoblastoma (RB) protein has also been shown to contribute to the post-translational regulation of ERα in BC cell lines, through the interaction between the N-terminal domain of RB and DBD, and the hinge region of ERα, the chaperone proteins HSP90 and p23 bind to ERα and protect it from UPS-induced degradation. In human and mouse BC cells, *RB* knock-down was associated with significantly lower ERα protein levels, thus suggesting a possible role of RB loss in the establishment of a ERα− phenotype [68].

While ERα polyubiquitination promotes its degradation via the proteasome, ERα monoubiquitination leads to ERα stabilization by preventing its polyubiquitination; the RNF31 protein, which associates to ERα in the cytoplasm of BC cells and promotes its monoubiquitination, is responsible for increased ERα levels and enhanced ERα activity [69].

Finally, ERα palmitoylation promotes ERα stabilization and its recruitment to the cell plasma membrane, where it activates oncogenic signaling through mechanisms that do not require a reprogramming of the gene expression. ERα mutations preventing palmitoylation were reported to be associated with ERα degradation [70].

### 2.6. The Role of Hypoxia

Hypoxia, defined as a suboptimal concentration of oxygen in the tumor microenvironment resulting from a combination of an excessive tumor growth rates and insufficient or aberrant tumor vascularization, has been crucially implicated in the stimulation of tumor cell growth, metabolic reprogramming, and metastasis [71]. Interestingly, one preclinical study showed that hypoxia reduces ERα protein levels, but not *ESR1* mRNA levels, in HR+ BC cell lines via the proteasome-dependent degradation of ERα and the HIF-1α-mediated repression of *ESR1* gene transcription (Figure 1); however, data linking ERα expression to HIF-1α signaling are still controversial [72,73,74].

### 2.7. The Role of BRCA1

Approximately 10–20% of BCs arising in patients with pathogenetic germline *BRCA1* mutations express ERα [75]. ERα+ BCs arising in *BRCA1*-mutated carriers typically show characteristics of biological and clinical aggressiveness, including a higher tumor grade and higher proliferation rates when compared with ERα+ BC occurring in *BRCA1* non-carriers [76]. In addition to its role in DNA damage response, BRCA1 is directly involved in regulating gene transcription in cancer cells, partly as a component of the RNA polymerase II holoenzyme complex via its interaction with RNA helicase A [77]. In the study by Hosey A.M. et al., wild-type BRCA1 was shown to directly activate the *ESR1* gene transcription by binding to the *ESR1* promoter and by recruiting Oct1 (Figure 1) [78]. As a consequence of the BRCA1-induced transcription of *ESR1*, several BCs arising in subjects carrying germline *BRCA1* inactivating mutations might display a low ERα expression as a result of the loss of the BRCA1-mediated transcription of *ESR1*. However, also in subjects with wild-type germline *BRCA1*, a progressive decrease of BRCA1 protein levels or reduced BRCA1 activation in BC cells as a result of genetic or epigenetic mechanisms, such as *BRCA1* gene LOH, *BRCA1* promoter hypermethylation, or transcriptional repression, might be responsible for reduced *ESR1* gene transcription and ERα protein levels [78]. Therefore, even during sporadic BC tumorigenesis, ERα expression levels might be progressively reduced in some patients as a result of a progressive reduction of BRCA1 expression/activity. Consistent with this hypothesis, *BRCA1* mRNA expression levels have been shown to positively correlate with *ESR1* mRNA levels in patients with sporadic BC, while reduced BRCA1 levels or activation, which can occur in sporadic BCs during tumor progression, or as a result of treatment-induced modifications of tumor biology, could directly impact ERα loss [79]. With regards to the PgR expression, wild type BRCA1regulates PgR levels by modulating the E3 ubiquitin ligase activity; in detail, BRCA1 promotes PgR protein ubiquitination and degradation, and, at the same time, targets the hormone-responsive regions (HRR) of the PgR target genes and induces chromatin silencing at PgR-regulated promoters via the BRCA1/BARD1 complex (Figure 2) [80].

### 2.8. Intratumor Heterogeneity

Loss of ERα and/or PgR expression in recurrent HR+ BC can result from the clonal selection of biologically heterogeneous primary or metastatic BC lesions during the course of ETs, i.e., under the selective pressure of reduced extracellular estrogens or the pharmacological inhibition of the ERα pathway (Figure 3) [81]. 

Intratumor heterogeneity exists in the presence of tumor cell clones with different ERα and/or PgR expression within the same tumor lesion. Several preclinical studies have investigated this phenomenon; Graham M. et al. validated a new flow cytometry-based immunoassay and associated software for the simultaneous quantification of PgR levels and DNA indices of ploidy and cell cycle stages in any subset of the total cell population using the human BC cell line T47D and its clonal derivatives, and demonstrated a remarkable heterogeneity in PgR expression, thus confirming the existence of distinct tumor cell subclones that show a mixed response to antiestrogen treatment with tamoxifen [82]. More recently, a new method for ER mature transcript quantification in single cells has been validated in a study by Annaratone L. et al., consisting in a single-molecule RNA fluorescent in Situ Hybridization (FISH) in formalin-fixed, paraffin-embedded tissue sections (FFPE-smFISH), which allows for the quantification and spatial localization of the heterogeneous intratumor ERα expression, thus revealing that its spatial distribution can vary substantially even in the same tumor lesion [83].

Remarkably, the presence of intratumor heterogeneity of ERα expression has been associated with an increased risk of patient death [84]. This may depend on the fact that heterogeneous tumors are characterized by a higher number of biologically heterogeneous tumor clones with an augmented capacity to adapt to unfavorable growth conditions, including the selective pressure of anticancer treatments [84]. In the work by Davis B.W. et al., multiple intratumor tissue samples from 32 BC patients were analyzed, and in 4 out of 32 patients (12.5%), both HR+ and HR− tumor areas were detected in the same lesion [85]. In the work of Chung G.G. et al., a hybrid IHC and flow cytometry technique employing fluorophores was used to analyze ERα expression in multiple tumor blocks/slides from primary BCs; notably, this study revealed a remarkable block-to-block heterogeneity in as much as 81% of cases [86]. Although the technique used in this study may have resulted in an over-estimation of the presence of intratumor heterogeneity, nevertheless, these results indicate that intratumor heterogeneity in the ERα expression is much more common than actually acknowledged. In a large series of 1085 invasive BC tissues, discordant biomarker status by tissue microarray was found in 9% and 16% of the analyzed samples for ERα and PgR, respectively, consistent with the existence of HR spatial heterogeneity [87].

Circulating tumor cells (CTCs), which detach from the primary tumor and move into the blood to establish metastatic seeds in distant organs, are detected in 10–67% of patients with any-stage BC, and are useful blood-based markers to predict clinical outcomes and to follow response to therapies [88]. Interestingly, as CTCs have similar chances to detach from different disease sites, they could potentially recapitulate intratumor heterogeneity. In the work of Babayan A. et al., among 35 patients with metastatic BC and ERα+ primary tumors, CTCs were detected and immunostained in 16/35 patients (46%). Notably, in only 31% of these cases did the CTCs show a homogeneous ERα expression, while in 50% of patients both ERα+ and ERα− CTCs were detected; finally, 19% of cases only showed ERα− CTCs [89].

## 3. HR Loss in Specific Clinical Settings

HR status is unstable during BC progression as a result of genetic and epigenetic mechanisms; pre-existing intratumor heterogeneity; and clonal cell selection under the pressure of antitumor therapies, including ETs and chemotherapy.

Importantly, the occurrence of a biological tumor switch from HR+ to HR− status is associated with important prognostic implications; in a retrospective analysis including 459 patients with available tissues from both primary BC and local/distant BC recurrences, patients with neoplasms undergoing a conversion from ERα+ to ERα− status had a 48% increase in the risk of death when compared with BC patients with stably ERα+ status (hazard ratio, 1.48; 95% CI, 1.08 to 2.05) [90].A loss of PgR expression in tumors that retain a high ERα expression has also been associated with more aggressive clinical behavior [33]. In particular, PgR loss promotes BC cell resistance to SERMs, but not to therapies that act by reducing blood estrogen concentration, such as aromatase inhibitors, or drugs that bind ERα and stimulate its degradation, such as SERDs [33]. It is unclear whether these data simply reflect a lower availability of effective therapies for ERα- BC when compared to ERα+ BC, or if they reflect the fact that cancer cells able to modify their epigenetic programs tend to display higher clinical aggressiveness.

In patients with early-stage BC treated with upfront surgery, the discordance rate in HR status between core needle biopsies (CNB) and paired surgical samples was reported to be low overall, i.e., 1.8% for ERα and 15% for PgR [91]. Similarly, discordances in ERα status between primary tumors and synchronous axillary lymph node metastases were found to be rare in a small patient cohort (2/50, 4%) [92]. These data indicate that HR status evolution does not occur in the short-time interval before surgery (i.e., in patients not treated with neoadjuvant therapies), and that intratumor heterogeneity, which is likely to explain most of the observed discrepancies between diagnostic tumor biopsies and surgical specimens, is an overall uncommon finding in early-stage BC lesions [91,93].

When investigating HR status conversion, some technical aspects regarding the methodology of HR assessment should be taken into account. Indeed, ERα and PgR status is assessed by means of IHC, which evaluates the proportion of tumor cells expressing ERα or PgR, and which dichotomizes BC cells as HR+ or HR− based on a cut-off point of 1% for both proteins [94]. Firstly, different sampling methods (fine needle aspiration vs. CNB vs. surgical resection of the whole tumor lesion) may contribute to discrepant results for the IHC assessment of HRs; in particular, higher levels of HR expression are found in CNB-samples when compared with surgical specimens, which, more commonly undergo delays in tumor fixation that may result in degradation of the thermolabile ERα or PgR and, consequently, in lower IHC staining of these proteins [95]. Secondly, the HR status definition is affected by the reproducibility of the IHC staining techniques [96]. However, the standardization of validated IHC techniques for HR assessment in recent years, along with the reduction of non-validated assays, has greatly improved the quality and reproducibility of ERα and PgR testing [94]. Particular challenges for test reproducibility are represented by BC specimens with a low ERα and/or low PgR expression (1–10%), whose assessment could be affected by both pre-analytic (e.g., fixation type and time) and analytic factors (e.g., antibody used), as well as by the tumor heterogeneity of the HR expression [94]. Finally, it should be highlighted that bone is the most frequent site of BC recurrences, and it has been reported that the decalcification process of biopsy samples, which is mandatory for IHC assessment in bone CNB specimens, may potentially alter the HR expression in BC [97]. 

The following paragraphs aim at discussing the clinical contexts in which HR loss has been reported in patients with BC.

### 3.1. HR Loss after Neoadjuvant Treatments

NAC is commonly used in BC patients with clinical stage II–III BC, for the purposes of (1) testing tumor sensitivity to systemic anticancer agents; (2) reducing tumor volume and, eventually, achieving pathologic complete response (pCR), which is associated with better long-term clinical outcomes; and (3) increasing the chance of performing conservative surgery. In recent years, NAC has become the standard-of-care treatment for patients with stage II–III TNBC, HER2+ BC, and luminal B-like BC, which are characterized by higher growth rates and more aggressive clinical behavior.

Some published studies indicate that HR status can be significantly modulated by neoadjuvant treatments [98,99,100]. Whether these changes are the result of the chemotherapy-induced selection of pre-existing tumor cell clones, or of chemotherapy-induced epigenetic modifications leading to changes in the ERα/PgR expression, remains unclear. Independently of the molecular origin, NAC-induced loss of HR expression can have clinically relevant consequences; indeed, while the assessment of HR and HER2 status in CNBs is essential to guide the choice of preoperative treatments, biological tumor re-characterization after systemic neoadjuvant treatment in those patients failing to achieve pCR is essential for guiding post-surgical treatment planning. For instance, post-neoadjuvant T-DM1 has recently demonstrated improved DFS and OS in HER2+ BC patients failing to achieve pCR during pre-operative trastuzumab-based biochemotherapy [101], while post-neoadjuvant capecitabine has proven to be effective in increasing DFS and OS in TNBC patients failing to achieve tumor pCR after neoadjuvant therapy [102]. Based on these studies, assessment of the HR and HER2 status in tumor specimens after NAC is crucial for the definition of the subsequent therapeutic strategies, and therapeutic indications might change when modifications in HR or HER2 status are detected in surgical specimens when compared with diagnostic CNBs.

Several large studies have compared the ERα/PgR expressions in primary tumors before and after NAC, with conflicting results, as illustrated below (Table 1). In an attempt to summarize these data, Van de Ven S. et al. conducted a meta-analysis of 32 significant studies that investigated the changes in ERα, PgR, and HER2 status in post- vs. pre-surgical BC samples in patients that received NAC or neoadjuvant trastuzumab-based biochemotherapy [103]. Studies employing neoadjuvant endocrine therapy were excluded from the meta-analysis. Overall, discordances in the HR status between the first CNB and post-NAC tumor specimens were reported in 8–33% of patients. In general, larger studies reported higher rates of HR status conversion, and also showed a more marked discordance in PgR status (discordance rates for ERα and PgR status were 2.5–17% and 5.9–51.7%, respectively). Interestingly, the meta-analysis revealed a correlation between the changes in HR and HER2 status after NAC. In particular, most tumors displaying an increase in ERα expression after neoadjuvant treatment also underwent a decrease in HER2 expression; similarly, loss of HER2 amplification was associated with an increased ERα expression after NAC plus trastuzumab-based treatment. Together, these data support the hypothesis that similar molecular pathways, or a common transcriptional instability, could sustain these biological switches. 

When considering individual trials included in the meta-analysis, the results regarding HR conversion were quite discordant, thus indicating that the centers in which the study was conducted, or the differences in patient characteristics, patient number, technique of HR status assessment, tumor sampling, and concomitant therapies administered, might significantly affect the rate of HR status conversion.

In a retrospective study including 420 patients with operable BC and treated with different NAC schedules, Tacca O. et al. reported changes in HR status in 23% of patients, with HR− to HR+ status conversion being the most commonly reported event (42% of initially HR− patients) [104]. In this study, patients undergoing a biological switch from HR− to HR+ status had significantly better DFS and OS when compared with patients with stable HR− status after NAC, while no significant differences in clinical outcomes were observed between patients undergoing HR− to HR+ status conversion and patients with stable HR+ tumors. These results indicate that the biological status of the tumor at surgery can affect patient prognosis and the clinical benefit from adjuvant therapies. 

However, in the study by Chen S. et al. [105], patients with HR+ to HR− conversion did not show a different clinical benefit from adjuvant ETs when compared with patients with a stable HR+ status, thus indicating that HR+ BC micrometastatic clones retain sensitivity to ETs in the vast majority of patients undergoing HR+ to HR− conversion. Loss of HR positivity was also associated with worse DFS and OS in the studies by Jin X. et al. [106] and Lim S.K. et al. [107]; finally, Ding Y. et al. found that a loss of PgR positivity was associated with worse OS [108]. 

Regarding the impact of neoadjuvant ETs on HR status, a few published studies indicate that preoperative tamoxifen or aromatase inhibitors could reduce the ERα and PgR expression, respectively [109,110]. The different effects of different ETs on the ERα and PgR expressions could result from the different mechanisms of action of these compounds. Indeed, while SERMs bind ERα and could promote its ubiquitination and degradation, aromatase inhibitors reduce ERα binding to its agonists, thus inhibiting ERα-mediated signaling without affecting ERα levels. In the outdated study by Hawkins R.A. et al., neoadjuvant tamoxifen caused a marked reduction in ERα expression, in contrast to most other forms of systemic therapy that only resulted in minor changes in Erα levels [111]. In the study by Miller W.R. et al., neoadjuvant tamoxifen caused a reduction in ERα expression along with an increase in PgR (likely as a consequence of estrogen agonistic activity of tamoxifen); on the other hand, letrozole had minor effects on ERα expression, but it was associated with a dramatic reduction of PgR levels, likely as a result of lower blood estrogen and lower ERα activation and signaling [112].

To the best of our knowledge, no studies evaluating HR conversion in the context of HR+ BC patients treated with neoadjuvant ET plus cyclin-dependent kinase (CDK) 4/6 inhibitors have been published yet.

Taken together, the available clinical evidence shows that HR conversion is a common finding in BC patients receiving NAC. Based on these data, HR status should be always re-assessed in surgical tumor specimens after NAC in order to select the most appropriate post-NAC systemic treatment.

### 3.2. HR Loss in Recurrent/Metastatic BC

In spite of continuous progresses in the treatment of early-stage BC during the last decades, BC recurrence, in the form of local or distant relapses, occurs in approximately 15–20% of patients with surgically-resected HR+/HER2− BC [113,114]. Therefore, recurrent BC remains an unmet clinical issue, and the vast majority of patients with metastatic BC still die of their disease.

In a retrospective study by Simmons C. et al., 20% of patients with metastatic BC underwent a change in their treatment plan based on the results of the biopsy of one metastatic lesion, and in particular, on the basis of HR and HER2 status re-assessment [115]. Therefore, the treatment of recurrent and metastatic BC should be guided by biological re-characterization of the tumor, which should include a re-assessment of the HR and HER2 status. Although current clinical guidelines recommend biological re-characterization of tumor recurrences as frequently as possible, the optimal frequency of HR re-assessment (i.e., after every change of systemic treatment vs. after the diagnosis of the first local/distant recurrence) remains unclear.

Changes in HR status between the primary tumor and metastatic lesions have prognostic value, with the conversion from HR+ BC to TNBC being associated with the worst clinical outcome [10]. Conversely, the acquisition of the HR expression in previously HR− BC is associated with better clinical outcomes, partly as a result of a higher number of available treatment options for patients with HR+ BC. 

The meta-analysis by Aurilio G. et al., which included 48studies, aimed at comparing HR status in primary BCs and in matched BC recurrences [15]. The biological features of primary BCs were reassessed at relapse to estimate the rates of ERα and/or PgR status conversion. Studies included in this meta-analysis were heterogeneous in terms of the treatments administered and the number of patients included. ERα status was assessed in 33 studies; with a total number of 4200 patients analyzed, the overall rate of ERα status conversion in primary vs. recurrent tumor specimens was 20% (95% CI: 16–35%); in particular, HR conversion from ERα-positive (ERα+) to ERα-negative (ERα−) status occurred in approximately 24% of patients (95% CI: 9–20%), while ERα− to ERα+ conversion occurred in 14% of patients (95% CI: 9–20%). PgR status conversion was assessed in 24 studies, with a total of 2739 patients analyzed, PgR status conversion was reported in approximately 33% of cases (95% CI: 28–37%). The pooled rates of PgR-positive (PgR+) to PgR-negative (PgR−) and PgR− to PgR+ conversion were 46% (95% CI: 37–55%) and 15% (95% CI: 12–17%), respectively (*p* < 0.0001). Interestingly, stratified analyses according to the site of the tumor relapse revealed similar pooled ERα discordance proportions in the case of loco-regional and metastatic recurrences, while discordance proportions for PgR expression were lower in loco-regional tumor recurrences (26%; 95% CI: 21–32%) than in distant metastases (41%; 95% CI: 37–45%). Notably, studies included in the meta-analysis showed discordant survival outcomes associated with HR conversion; for instance, while, in the study by Amir E. et al., patients undergoing HR status conversion did not have worse OS, in the study by Dieci M.V.et al., HR+ to HR− conversion was associated with significantly lower post-relapse OS [116,117]. Based on the results of this meta-analysis, the authors highlighted the importance of re-assessing the tumor biology, especially in the following clinical contexts: (1) BC recurrences occurring along time after primary BC diagnosis (which indicates the evolution of tumor cell clones that remained quiescent for a long time, and which probably underwent progressive clonal selection during/after adjuvant therapies), (2) disease progression in the context of early and frequent treatment failures (which suggest the presence of highly dynamic tumor cell clones that could have been rapidly selected during previous therapies), and (3) conditions where the detection of a specific biological alteration could guide clinicians in the choice of molecular targeted therapies. Whether HR conversion from primary BC to recurrent/metastatic BC occurs more frequently after the administration of post-surgical systemic treatments has been poorly investigated (Table 2).

In the retrospective study by Kuukasjärvi T. et al., which included 50 patients not receiving any adjuvant treatment, the discordance rate in HR status between primary BC samples and matched metachronous BC recurrences was 36%; in this study, all conversion events consisted in the loss of ERα and/or PgR expression, while no ERα− to ERα+ or PgR− to PgR+ conversions were detected [81]. In this study, a loss of Erα expression in recurrent BC was associated with a poor clinical benefit from ETs, with only 12.5% of patients with ERα− tumors responding to tamoxifen-based therapy (compared with 74% of patients retaining intratumor Erα expression). Lower E.E. et al. retrospectively compared HR expression in matched primary and metastatic BC specimens from 200 patients, 167 of whom received adjuvant treatments, which consisted of adjuvant chemotherapy alone in 102 patients, or of tamoxifen, alone or with other therapies, in 65 patients [118]. The authors found a discordance rate of 30% regarding ERα status, with a switch from ERα+ to ERα− status in19.5% of patients. The discordance rate was 39.3% for the PgR expression. In this study, adjuvant tamoxifen was not specifically associated with increased ERα discordance rates; indeed, ERα expression was lost in 34% of patients who were treated with tamoxifen and in 38% of patients who were not. Of note, patients with tumors retaining the ERα expression at the time of recurrence had a significantly longer OS when compared with patients with tumors losing ERα expression; conversely, discordance in the PgR expression was not associated with patient OS.

In the study by Stueber T. et al., which included196 patients, the observed rate of ERα+ to ERα− conversion was 33.3%, while the PgR+ to PgR- conversion rate was 59.6% [119]. Notably, both tamoxifen and chemotherapy, when administered in the adjuvant setting, were associated with a significantly lower risk of undergoing disease recurrence as ERα− BC, thus indicating that in some contexts, ERα−/PgR− clones might originate from HR+ BC clones that undergo modifications in their epigenetic programs, rather than from the selection of initially ERα−/PgR− tumor clones. If this hypothesis is correct, adjuvant ETs could help prevent the formation of ERα− cell clones, rather than selecting them. In this paper, the authors also reported a higher risk of HR conversion at relapse in women who underwent axillary lymph node dissection because of initial lymph node tumor involvement, with a trend towards a higher conversion risk in the case of low-grade primary tumors. Conversely, no significant correlation between menopausal status or familial predisposition and any receptor change was found.

In contrast with previously discussed works [118,119,120], in the study by Lindström L.S. et al., ERα expression loss at recurrence was more common among patients who received ET alone (29%) or chemotherapy followed by ET (34.3%) in the adjuvant setting, while it was less frequent in patients treated with chemotherapy alone (19.8%) and in patients who did not receive any adjuvant therapy (11.5%) [90]. In contrast with the study by Stueber T., this study indicates that ETs could favor the formation HR− tumor clones, or simply select the growth and proliferation of pre-existing HR- subclones.

With regards to specific (neo)adjuvant treatment agents, in the study by Ongaro E. et al. [120], previous exposure to anthracyclines or taxanes was associated with a loss of PgR expression (OR 6.5) or with its quantitative reduction (OR 5.3). Conversely, previous exposure to taxanes or aromatase inhibitors was associated with a quantitative reduction of Erα expression in BC recurrences (OR of 3.6 and 2.5, respectively). As for the different metastatic sites, in the meta-analysis conducted by Schrijver W.A.M.E. et al. [16], the authors analyzed location-specific discordance in HR status among different studies, and found that the ERα expression status was more frequently discordant in the central nervous system (20.8%) and in bone metastases (29.3%) when compared with liver metastases (14.3%). On the other hand, the PgR status was more commonly discordant in the bone (42.7%) and liver metastases (47%) when compared with the CNS metastases (23.3%).

As in the context of neoadjuvant therapies, data on HR conversion in patients treated with ET plus CDK4/6 inhibitors in the setting of advanced disease are still lacking.

**Table 2 cells-09-02644-t002:** Previous studies investigating the discordance rate between primary tumor and local recurrence/distant metastases.

Author	Methods	Site	Previous Treatments	ERα (%)	PgR (%)	HR Status (%)
Kuukasjärvi T. et al. (1996) [81]	Retrospective N = 50 pts	Local recurrence/Distant metastases	None	24	24	36
Lower E.E. et al. (2005) [118]	Retrospective N = 200 pts	Distant metastases	Tamoxifen only/chemotherapy only/chemotherapy+ tamoxifen	30	39.3	-
Simmons C. et al. (2009) [115]	Prospective N = 25 pts	Distant metastases	Various	-	-	40
Liedtke C. et al. (2009) [121]	Retrospective N = 789 pts	Local recurrence/ Distant metastases	Taxanes vs. endocrine therapy	18.4	40.3	14–40
Thompson A.M. et al. (2010) [122]	Prosepective N = 137 pts	Local recurrence/ Distant metastases	Various	10	25	15
Amir E. et al. (2012) [116]	Prospective N = 117 pts	Distant metastases	Various	16	40	37.6
Ongaro E. et al. (2018) [120]	Retrospective N = 232 pts	Distant metastases	Anthracyclines (50.4%); taxanes (33.6%); antiestrogens (49.1%); AI (39.2%)	12.7	49.7	-
Schrijver W.A.M.E. et al. (2018) [16]	Metanalysis N = 39 studies	Distant metastases	Various	19.3	30.9	-
Stueber T. et al. (2019) [119]	Retrospective N = 196 pts	Local recurrence/ Distant metastases	Various	36.8	75.4	-

## 4. Therapeutic Implications

As HR status conversion is a common phenomenon in BC patients treated with different types of antineoplastic therapies, and residual tumor tissues after NAC, local or distant BC recurrences after the resection of the primary tumor, and tumor sites undergoing progression during subsequent lines of therapy for advanced disease should be biologically re-assessed whenever possible so as to re-evaluate HR status. In this clinical context, the re-characterization of both HR and HER2 status is crucial in order to guide the choice of subsequent lines of therapy. 

In post-NAC tumor residues, changes in HR and/or HER2 status could significantly modify therapeutic decisions regarding post-neoadjuvant therapies. For instance, patients with TNBC undergoing conversion towards HR+ BC could benefit from adding ETs to adjuvant capecitabine, while tumor conversion from HR+ BC to TNBC after NAC might suggest an advantage from adding capecitabine to adjuvant ETs. Similarly, tumor conversion from HER2+/HR− BC to HER2+/HR+ BC suggests the opportunity to combine ETs with post-neoadjuvant T-DM1 or trastuzumab, while tumor conversion from HER2−/HR+ BC to HER2+/HR+ BC indicates the potential utility of adding T-DM1 to adjuvant ETs. As a general principle, combination therapies targeting both HR+ and HR− BC cell clones in patients with tumors undergoing HR status conversions should be preferred whenever possible. Indeed, HR status conversion suggests the potential presence of both HR+ and HR− BC cell clones that could be responsible for the long-term formation of distant metastatic lesions.

In the case of distant BC recurrences after (neo)adjuvant therapies or surgery for limited-stage BC, clinical guidelines aimed at biological tumor re-characterization indicate soft tissue BC biopsies as a preferred option when compared with biopsies of bone metastases, because the former allow for more accurate biomarker assessment and preserve tumor specimens from alterations resulting from the use of harsh decalcification solutions [11]. Whenever the occurrence of HR status conversion in BC recurrences is confirmed, therapeutic decisions should be customized on a case-by-case basis, i.e., depending on the burden of the disease, the number and type of previous lines of therapy received, tumor response to these therapies, and the duration of the response. At the extreme of the clinical spectrum, if tumor relapse after curative surgery occurs in a single distant site and ERα− to ERα+ conversion is detected, first-line ET plus/minus CDK 4/6 inhibitors should be the preferred treatment option. Conversely, if the patient presents with multiple metastatic sites from a previous ERα− BC, and there is proven ERα positivity in one biopsied site, ET plus CDK 4/6 inhibitor combination could be proposed as a first-line therapeutic option (with the exception of ongoing or pending visceral crisis), but then it should be replaced by cytotoxic chemotherapy or other treatment options if one or more metastatic sites undergo progression at disease re-evaluation. In contrast, in the case of ERα+ to ERα− conversion, treatment choices should be guided by the clinical course of the disease. In the presence of one single metastatic lesion or oligometastatic disease, cytotoxic chemotherapy could be the preferred treatment option. In the case of multiple metastatic sites, combining chemotherapy with ETs could increase the chances of successfully targeting biologically heterogeneous (ERα+ and ERα−) tumor lesions.

In the future, new methods capable of reliably detecting the heterogeneity of HR status in different metastatic tumor lesions could revolutionize the therapeutic algorithm in patients undergoing HR− to HR+ or HR+ to HR− tumor conversions. In a recent study by Paolillo C. et al., which was conducted in patients with advanced HR+ BC treated with ET, ERα expression in CTCs revealed a remarkable heterogeneity across different cells. In particular, most samples in which CTCs were detected showed a biologically mixed population of ERα+ and ERα− BC cells, and in five patients, all CTCs were negative for ERα expression [123]. While the assessment of HR status in CTCs is a promising approach that could allow for the detection of HR heterogeneity in BC patients, a relationship between ERα expression in CTCs and the presence of specific metastatic sites have not been elucidated yet. In particular, it is still unclear if CTCs equally represent biologically heterogeneous metastatic lesions, or, alternatively, if they represent the most biologically and clinically aggressive tumor clones (e.g., those lacking ERα expression). 

Another innovative approach consists of the development and improvement of imaging techniques capable of providing in vivo visualization of ERα expression in individual tumor lesions. Among these methods, 16a-[18F]-fluoro-17b-estradiol (FES) PET/CT is capable of accurately localizing tumor lesions expressing ERα, and also of predicting ET sensitivity in patients with advanced HR+ BC [124]. Of note, FES PET/CT scans performed before the initiation of ETs in patients with advanced ERα+ BC revealed heterogeneous FES avidity across different individuals [125,126]. In the study by Linden H. et al., the lack of FES uptake was reported in at least one metastatic site in 10% of patients with primary ERα+ BC, reflecting the presence of different tumor clones [127]. Conversely, the reduction in FES avidity during ETs may be due to receptor occupancy and/or loss. Repeating FES PET/CT scans during ET course could indeed guide treatment choices. 

While new diagnostic approaches could increase our ability to visualize HR status heterogeneity in HR+ BC and, consequently, to guide treatment choices in a personalized way, prospective studies are needed in order to investigate whether tailoring pharmacological treatments in HR+ BC patients on the basis of intratumor heterogeneity of HR expression is capable of improving clinical outcomes when compared with the currently available clinical guidelines.

## 5. Conclusions

HR+ to HR− conversion is a common biological phenomenon in BC patients, especially after neoadjuvant therapies (i.e., in residual tumor cells) and in the case of BC recurrences after surgery. Regardless of the underlying molecular mechanisms, which include treatment-driven genetic/epigenetic alterations affecting *ESR1/PGR* transcription, *ESR1/PGR* translation, or ERα/PgR activity, as well as intratumor HR status heterogeneity, this phenomenon can severely impact treatment choices and the efficacy of antitumor therapies. Therefore, performing tumor re-biopsy to achieve HR status re-characterization of BC recurrences or progressing lesions should be recommended whenever feasible in order to tailor treatment on the basis of HR status. At the same time, as different tumor lesions can display different HR expressions, new methods allowing for the visualization of heterogeneous tumor lesions, such as HR characterization in CTCs or FES PET/CT scans, could help clinicians choose the most effective treatment in specific clinical contexts.

## Figures and Tables

**Figure 1 cells-09-02644-f001:**
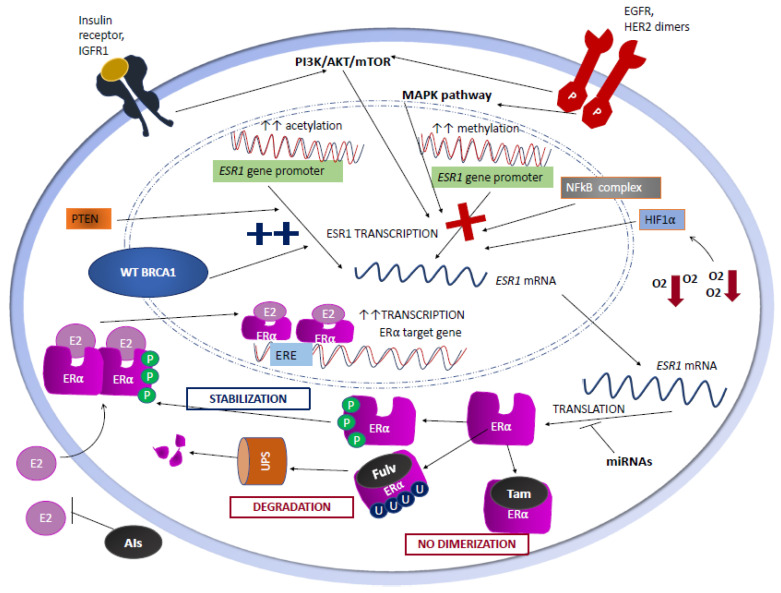
Mechanisms that regulate *ESR1* transcription, *ESR1* mRNA translation, and ERα post-translational modifications in HR+ BC cells. BRCA1: Breast Cancer Type 1 susceptibility protein; E2: estradiol; ERα: estrogen receptor alpha; ERE: estrogen responsive element; AIs: aromatase inhibitors; Fulv: fulvestrant; miRNAs: microRNAs; P: phosphate group; PTEN: phosphatase and tensin homolog; Tam: tamoxifen; U: ubiquitin; UPS: ubiquitin proteasome system.

**Figure 2 cells-09-02644-f002:**
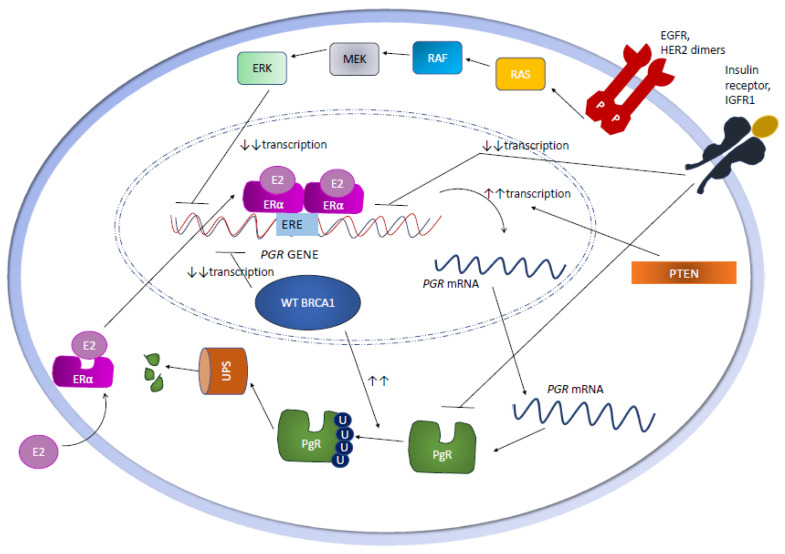
Mechanisms that regulate *PGR* transcription, *PGR* mRNA translation and PgR post-translational modifications in HR+ BC cells. BRCA1: Breast Cancer Type 1 susceptibility protein; E2: estradiol; ERα: estrogen receptor alpha; ERE: estrogen responsive element; PTEN: Phosphatase and tensin homolog; U: ubiquitin; UPS: ubiquitin proteasome system; WT: wild type.

**Figure 3 cells-09-02644-f003:**
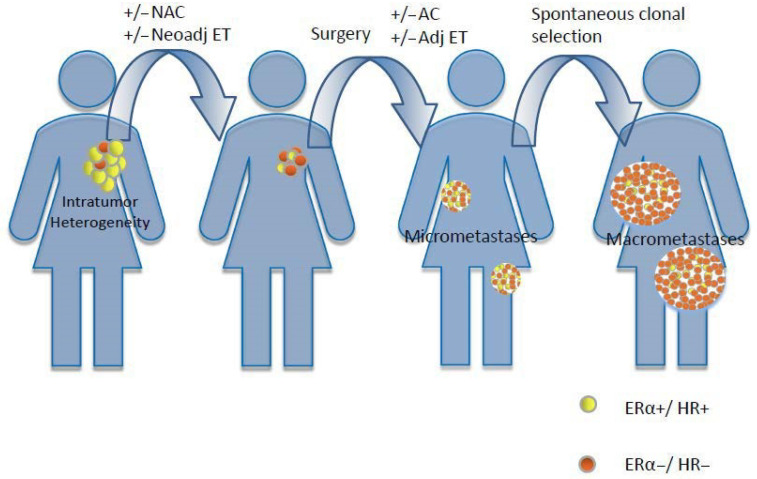
Presence of baseline tumor heterogeneity, with the coexistence of ERα+/HR+ and ERα−/HR− clones and selective pressure of different treatments, which lead to the progressive selection of tumor lesions enriched for ERα−/HR− cells. NAC: neoadjuvant chemotherapy; Neoadj ET: neoadjuvant endocrine therapy; AC: adjuvant chemotherapy; Adj ET: adjuvant endocrine therapy.

**Table 1 cells-09-02644-t001:** Studies investigating the discordance rate between core needle biopsies (CNB)and synchronous LN (lymph nodes)/surgical specimens.

Author	Methods	Site	Previous Treatments	ERα (%)	PgR (%)	HR Status (%)
Hawkins R.A. et al. (1990) [111]	Prospective N = 62 pts	Surgical sample	Various	n.s	-	-
Miller W.R. et al. (2003) [112]	Retrospective N = 48 pts	Surgical sample	Letrozole vs tamoxifen	48 vs 83	87 vs 60	-
Tacca O. et al. (2007) [104]	Retrospective N = 459 pts	Surgical sample	Various	-	-	23
Hirata T. et al. (2009) [99]	Retrospective N = 368 pts	Surgical sample	Various	14.9	29.1	16
Aitken S.J.et al. (2010) [93]	Retrospective N = 385 pts	Synchronous LN	None	28.3	23.4	-
Van de VenS. et al. (2011) [103]	Metanalysis N = 32 studies	Surgical sample	Various	2.5–17	5.9–19	8–33
Jensen J.D. et al. (2012) [92]	Prospective N = 128 pts	Synchronous LN	None	4	-	-
Chen S. et al. (2012) [105]	Retrospective N = 224 pts	Surgical sample	CEF (29.0%), NE (35.7%), PC (29.9%), or TE (5.4%) ^1^	16 (+ to −)	22.2 (+ to −)	15.2 (+ to −)
Jin X. et al. (2015) [106]	Prospective N = 423 pts	Surgical sample	Various	-	-	18.4
Lim S.K. et al. (2016) [107]	Retrospective N = 322 pts	Surgical sample	Various	40.7	62.1	17.9
Ding Y. et al. (2020) [108]	Retrospective N = 482 pts	Surgical sample	Anthracyclines and/or taxanes	10.4	17	27.4

^1^ CEF: cyclofosphamide + epirubicin + 5-fluorouracil; NE: navelbine + epirubicin; PC: paclitaxel + carboplatin; TE: docetaxel + epirubicin.

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
