# Peer review of "Hormone Receptor Loss in Breast Cancer: Molecular Mechanisms, Clinical Settings, and Therapeutic Implications"

_cells, 2020, doi:10.3390/cells9122644_

Round 1
Reviewer 1 Report
This manuscript by Zattarin et al. This informative manuscript summarizes hormone receptor positive to negative conversion in breast cancer cases.
The underlying mechanisms of this conversion are clearly described.
The paper's concept is interesting, however, there are several concerns for the publication in the Cells.
Major comments
1. Overall, the text is so long that it is a little bit difficult for me to understand the core of this manuscript. It might be better to pick up some important points and shorten the rest manuscript so that the readers can easily understand what the authors want to tell, especially clinical section should be summarized.
2. In Figure 1, please note what objects represent ,for example, quad phosphates with red circle, the object binding with E2. The authors should remake figure 1 following the molecular mechanisms of the loss of hormone receptors. I suggest that the authors might make a figure for each section (section:2.1-2.8) in the manuscript.
And the authors should add the overview of PgR regulation in the figure.
Miner comments
1. In line 36 of page 1, please add references that show ERβ reduces the effect of ERα-induced stimulation of cell proliferation.
2. In line 84 of page 2, please include references concerning the modulation of HR expression by cytotoxic agents.
3. In line 141-144 of page 3-4, citation is required as to PGR gene methylation.
4. In Table 1 and 2, it might be better to place the studies in a chronological order.
Author Response
Reviewer n.1
This manuscript by Zattarin et al. This informative manuscript summarizes hormone receptor positive to negative conversion in breast cancer cases.
The underlying mechanisms of this conversion are clearly described.
The paper's concept is interesting, however, there are several concerns for the publication in the Cells.
We thank the reviewer for finding our manuscript informative, and for the overall positive comments about our work.
Major comments
- Overall, the text is so long that it is a little bit difficult for me to understand the core of this manuscript. It might be better to pick up some important points and shorten the rest manuscript so that the readers can easily understand what the authors want to tell, especially clinical section should be summarized.
As suggested by the referee, we shortened the length of the manuscript, in particular with regards to the clinical section of the paper. In doing so, we maintained the same structure and paragraph organization, but we summarized some clinical studies. The new version of the submitted manuscript contains 18 pages, including text, figures and tables and excluding references. We thank the reviewer for this suggestion, which in our opinion contributed to make the paper clearer for the reader.
- In Figure 1, please note what objects represent ,for example, quad phosphates with red circle, the object binding with E2. The authors should remake figure 1 following the molecular mechanisms of the loss of hormone receptors. I suggest that the authors might make a figure for each section (section:2.1-2.8) in the manuscript.
And the authors should add the overview of PgR regulation in the figure.
We modified Figure 1 to make it clearer. In particular, we clarified the meaning of each symbol indicated in the Figure. We also included an additional figure (Figure 2) illustrating potential mechanisms leading to PgR down-regulation in HR+ BCs. Finally, we added a Figure 3, in which we illustrated how intratumor heterogeneity of HR expression could be responsible for the loss of HR expression in the metastatic tumor when compared to the primary one. We thank the reviewer for this suggestion.
Miner comments
- In line 36 of page 1, please add references that show ERβ reduces the effect of ERα-induced stimulation of cell proliferation.
As suggested by the referee, we added references describing the effect of ERβ in counteracting ERα-mediated stimulation of cell proliferation. We thank the reviewer for this suggestion.
- In line 84 of page 2, please include references concerning the modulation of HR expression by cytotoxic agents.
As suggested by the reviewer, we included references describing the effect of cytotoxic chemotherapy in modulating the expression of hormone receptors. We thank the reviewer for this suggestion.
- In line 141-144 of page 3-4, citation is required as to PGR gene methylation.
We added the required references describing the effects of PGR gene methylation on the expression of PGR.
- In Table 1 and 2, it might be better to place the studies in a chronological order.
As suggested by the reviewer, we modified the order of the studies presented in Tables 1 and 2 based on the chronological order of study publication. We thank the reviewer for this suggestion.
Reviewer 2 Report
This article is a well written analysis of factors influencing ESR1 expression and its implications for cancer prognosis. It is publication ready, with two minor comments.
- Line 94. ER is predominantly nuclear. PMID: 2683797, PMID: 2939725
- Line 209. They should mention that SERD activity likely does not drive antagonism PMID: 31353221, PMID: 21501600
Author Response
Reviewer n.2
This article is a well written analysis of factors influencing ESR1 expression and its implications for cancer prognosis. It is publication ready, with two minor comments.
We thank the reviewer for the positive comments about our manuscript.
- Line 94. ER is predominantly nuclear. PMID: 2683797, PMID: 2939725
We added the references indicated by the reviewer. We thank the reviewer for these suggestions.
- Line 209. They should mention that SERD activity likely does not drive antagonism PMID: 31353221, PMID: 21501600
As suggested, we added references to clarify the molecular mechanism of SERDs, and in particular to clarify that the activity of SERDS does not drive antagonism. We thank the reviewer for suggesting us to clarify this concept, and for indicating references in support of this concept.
Round 2
Reviewer 1 Report
The authors have addressed all my concerns, no further comments.